# SARS-CoV-2 Infection and Emery-Dreifuss Syndrome in a Young Patient with a Family History of Dilated Cardiomyopathy

**DOI:** 10.3390/genes12071070

**Published:** 2021-07-14

**Authors:** Irina Magdalena Dumitru, Nicoleta Dorina Vlad, Sorin Rugina, Nicoleta Onofrei, Sabina Gherca, Marian Raduna, Aurel Trana, Mirela Dumitrascu, Elena Popovici, Mircea Bajdechi, Lucia Zekra, Roxana Carmen Cernat

**Affiliations:** 1Clinical Infectious Diseases Hospital, Street 100 Ferdinand, 900709 Constanta, Romania; dumitrui@hotmail.com (I.M.D.); sorinrugina@yahoo.com (S.R.); mireladumitrascu33@gmail.com (M.D.); luciazekra@yahoo.com (L.Z.); roxana.cernat@seanet.ro (R.C.C.); 2Faculty of Medicine, Ovidius University of Constanta, Universității Street, nr. 1, B Building, 900470 Constanta, Romania; 3Doctoral School of Medicine, Ovidius University of Constanta, Universității Street, nr. 1, B Building, 900470 Constanta, Romania; resurse_umane@smuct.ro (A.T.); g.elenapopovici@gmail.com (E.P.); mircea.bajdechi@gmail.com (M.B.); 4Romanian Academy of Scientists, Spl. Independentei, 54, Bucharest-Sector 5, Bucharest 50085, 030167 Bucharest, Romania; 5Romanian Academy of Medical Sciences, Ion C. Brătianu Street, nr. 1, 030167 Bucharest, Romania; 6Medgidia Municipal Hospital, Strada Ion Creangă 18, 905600 Medgidia, Romania; onofreinico@yahoo.com; 7County Emergency Clinical Hospital “Sf. Apostol Andrei”, Tomis Street, nr. 145, 900591 Constanța, Romania; sabinagherca@ymail.com; 8Military Emergency Hospital “Dr. Alexandru Gafencu”, Mamaia Street, nr. 96, 900527 Constanța, Romania; rdn_marian@yahoo.com

**Keywords:** SARS-CoV-2, high-sensitive cardiac troponin I, myocarditis, Emery–Dreifuss syndrome

## Abstract

Emery–Dreifuss muscular dystrophy (EDMD) is a rare genetic disease that affects the musculoskeletal system, including the heart, causing rhythm disorders and cardiomyopathy, sometimes requiring an implantable cardioverter-defibrillator (ICD) or heart transplantation due to severe heart damage. The case described herein concerns a 16-year-old girl, with grade II obesity, without other known pathological antecedents or cardiac pathology diagnosis given an annual history of cardiological investigations. She was admitted to the Infectious Diseases Department with SARS-CoV-2 virus infection. The anamnesis showed that the cardiological investigations performed in the past were completed due to the medical history antecedents of her sister, who had been diagnosed with dilated cardiomyopathy, having undergone the placement of an ICD and a heart transplant. Numerous investigations were performed during hospitalization, which revealed high levels of high-sensitive cardiac troponin I (hs-cTnI), creatine kinase (CK) and N-terminal pro b-type natriuretic peptide (NT-proBNP). Dynamic electrocardiographic evaluations showed ventricular extrasystoles, without clinical manifestations. The patient presented stage 2 arterial hypertension (AHT) during hospitalization. A cardiac ultrasound was also performed, which revealed suspected mild subacute viral myocarditis with cardiomyopathy, and antihypertensive medication was initiated. A heart MRI was performed, and the patient was diagnosed with dilated cardiomyopathy, refuting the suspicion of viral subacute myocarditis. After discharge, as the patient developed gait disorders with an impossible heel strike upon walking and limitation of the extension of the arms and ankles, was hospitalized in the Neurology Department. Electrocardiograms (ECGs) were dynamically performed, and because the rhythm disorders persisted, the patient was transferred to the Cardiology Department. On Holter monitoring, non-sustained ventricular tachycardia (NSVT) was detected, so antiarrhythmic treatment was initiated, and placement of an ICD was subsequently decided and was diagnosed with EDMD. Genetic tests were also performed, and a mutation of the lamin A/C gene was detected (*LMNA* gene exon 2, variant c448A > C (*p.Thr150pro*), heterozygous form, AD).

## 1. Introduction

SARS-CoV-2 virus infection may cause viral myocarditis, with the literature claiming that the SARS-CoV-2 virus is a trigger for myocarditis, but there are no clinical data to support the link between the SARS-CoV-2 virus and the unfavorable evolution of the Emery–Dreifuss syndrome (EDMD) [1,2].

Emery–Dreifuss syndrome is a rare genetic disease, caused by mutations of emerin (*EMD*) and lamin A/C (*LMNA*) genes, and can be autosomal dominant or recessive [3,4]. The lamin gene has two types of mutation: type B and type A/C. The A/C lamin is encoded by the *LMNA* gene [4]. For patients with *EMD* and negative *LMNA*, studies also incriminate the *FHLA1* gene, which is associated with the *EDMD6* phenotype, which is X-linked and includes hypertrophic cardiomyopathy [3,4].

*EMD* mutation causes Emery–Dreifuss syndrome 1 (EDMD1), which is an X-linked recessive disease [4].

*LMNA* mutations are associated with autosomal dominant and autosomal recessive Emery–Dreifuss muscular dystrophy, Hutchinson–Gilford progeria syndrome limb girdle muscular dystrophy 1B, dilated cardiomyopathy with conduction defects, Charcot–Marie–Tooth axonal neuropathy type 2, familial partial lipodystrophy, mandibuloacral dysplasia, and atypical forms of Werner syndrome [4,5,6].

There are 41 known *LMNA* mutations, as follows: 23 of them cause autosomal dominant Emery–Dreifuss muscular dystrophy (EDMD2), 8 of them cause dilated cardiomyopathy and 1 mutation causes autosomal recessive Emery Dreifuss (EDMD3) [7,8].

*LMNA*-related dilated cardiomyopathy is characterized by left ventricular enlargement, reduced systolic function and arrhythmias, having unfavorable prognosis, with risk of sudden death [9].

EDMD is characterized by a clinical triad including muscle–joint illness, muscle weakness and cardiomyopathy with rhythm disorders due to a structural or functional defect of the genes encoding the proteins of the nuclear envelope, e.g., *LMNA*, this being a rare condition, with estimated incidence of 3:1,000,000 [4,10].

## 2. Case Presentation

This clinical case is of a 16-year-old female patient with unknown personal pathological history, with a cardiac hereditary history. Her sister was diagnosed with dilated cardiomyopathy, requiring ICD and heart transplant. The patient was hospitalized for SARS-CoV-2 virus infection, with mild COVID-19 symptoms including rhinorrhea and a dry cough.

At hospitalization, the clinical examination revealed a good general condition, with a temperature of 37.4 °C, SaO_2_ 99% at room temperature, rhythmic heart sounds, without retrosternal pain, tachycardia, or tachypnea. The patient presented stage 2 hypertension (blood pressure of 150/90 mmHg,), a heart rate of 80 beats per minute, normal ECG, without signs of meningeal irritation, and the Glasgow Coma Scale score was 15/15.

The clinical investigation results at admission were as follows:-Elevated values: high-sensitive cardiac troponin I (hs-cTnI 40.8 ng/L, normal < 6 ng/L); creatine kinase (CK 590 U/L, normal values < 192 U/L); N-terminal probrain natriuretic peptide (NT-proBNP 154 pg/mL, normal values < 125 pg/mL); D-dimers (580 ng/mL, normal values < 500 ng/mL); C-reactive protein (PCR 2 mg/dL, normal values 0.5 mg/dL); aspartate aminotransferase (AST 35 U/L, normal values < 25 U/L); alanine transaminase (ALT 50 U/L, normal values < 25 U/L), lactate dehydrogenase (LDH 248 U/L, normal values < 220 U/L);-Normal values: leukocytes (6.37 × 10^9^ cells/L), lymphocytes (2.37 × 10^9^ cells/L), neutrophils 6.2 × 10^9^ cells/L, platelets 360 × 10^9^ cells/L, creatinine 0.56 mg/dL, sodium 135 mmol/L, potassium 4.2 mmol/L.

Dynamic ECGs were performed on the 4th day of hospitalization and revealed ventricular extrasystole, and therefore a cardiologic examination was performed.

Heart ultrasound revealed a slightly dilated left ventricle, while the left ventricular ejection fraction (LVEF) was 55–60%, the left ventricle (LV) dimensions were 53/35 mm, with hypokinesia in the lower wall, a suspicion of viral subacute myocarditis and cardiomyopathy. Thus, conversion enzyme inhibitor treatment and heart MRI were performed.

During hospitalization, an antiviral treatment with lopinavir/ritonavir (400/100 mg q12h × 3 days) was initiated, and the patient was also prescribed anti-inflammatory medication (paracetamol 500 mg po bd × 7 days). An anticoagulant with prophylactic enoxaparin sodium (enoxaparin sodium 0.4 mL qd × 8 days), an antihypertensive with ACE inhibitors (Ramipril 2.5 mg po qd × 3 days) and a hepatoprotective drug (Silymarin 150 mg po td × 8 days) were also introduced. The antiviral treatment was stopped at the indication of the cardiologist after 3 days of administration.

The heart MRI performed immediately after discharge refutes the suspicion of viral subacute myocarditis and confirms the diagnosis of dilated cardiomyopathy (Figure 1).

Early gadolinium enhancement had a normal appearance, without evidence of intracavitary thrombosis or microvascular obstruction.

The late gadolinium enhancement revealed a small contrast outlet with mid-myocardial topography at the level of the basal segments of the inter-ventricular septum, respectively, and a small focal contrast outlet at the lower point of insertion in the right ventricle on the left ventricle.

T2 mapping: T2 value 49–50 ms (does not suggest myocardial edema).

T1 mapping: T1 values were slightly increased, without highlighting the expansion of the extracellular compartment. (T1 septal precontrast 1019 ms, ECV 24%).

The LV was slightly dilated (VTDi 98 mL/m^2^—indexed value was probably underestimated due to the patient’s weight), with systolic function around the lower limit of the normal value (LVEF 58%).

There was also highlighted a small area of focal mid-myocardial fibrosis at the basal interventricular septum and non-specific focal fibrosis at the lower insertion point of the RV on the LV.

The MRI diagnosed a slight dilation of the left ventricle, currently with preserved bi-ventricular systolic function, but with a small area of mid-myocardial focal fibrosis in the basal interventricular septum.

One month after discharge, the patient presented musculoskeletal system disorders, with gait disorders (with the ability to walk on her toes but impossible heel strike), Achilles’ tendon retractions and limitation of the extension of the arms and ankles but with present symmetrical deep tendon reflexes. Therefore, she was admitted to the Neurology Department and genetic tests were performed. The heterozygous mutation c.448A > C (*p.Thr150Pro*), in exon 2 of the *LMNA* gene was identified, confirming the diagnosis of Emery-Dreifuss Muscular Dystrophy.

She was transferred to the Cardiology Department due to rhythm disorders, as diagnosed by ECG, where a heart ultrasound and Holter monitoring were performed. The patient did not present with any clinical symptoms.

The Holter monitoring indicated ventricular events, representing less than 1% of the total events. It is important to specify the fact that the first-degree atrioventricular (AV) block, non-sustained ventricular tachycardia (NSVT) and isolated and systematized monomorphic extra-systolic ventricular arrhythmia in episodes of ventricular bigeminy were present throughout the recording (Figure 2).

Cardiac ultrasound revealed situs solitus, levocardia, normal atrio-ventricular connections, and normal pulmonary venous return (PVR), intact interatrial septum (IAS) and interventricular septum (IVS). A mild LV contractile dysfunction was observed, with LVEF of 53% and 0-I scale mitral regurgitation.

The heart ultrasound values were the following: interventricular septum (IVS) = 8 mm, left ventricle posterior wall thickness (LVPWs) = 9 mm, right ventricular end-diastolic volume (RVEDV) = 54 mm, left ventricular end-diastolic volume (LVEDV) = 39 mm, left atrium (LA) = 34 mm, right ventricle (RV) = 25 mm, right atrium (RA) = 36 mm, and left ventricular ejection fraction (LVEF) = 53%.

Thus, antiarrhythmic treatment with amiodarone was initiated during the treatment; the PR interval was prolonged, so it was decided to transfer the patient to the Institute of Cardiovascular Diseases and Transplantation in Târgu Mureş, a center specializing in ICD placement in children. In the absence of specific guidelines, it was decided by the multidisciplinary team that the benefits outweighed the risks, and the patient was scheduled for the implantation of a bicameral cardiac defibrillator.

After ICD placement, ECG was performed.

## 3. Discussion

Studies show that chronic diseases, including muscular dystrophies, are risk factors for the development of severe cases of SARS-CoV2 virus infection [4]. The patient in this case had a mild form of COVID-19.

The literature has not shown that there is an increased risk of rapidly unfavorable evolution of Emery–Dreifuss syndrome. Studies show that the severity of the disease depends on the genes expressed—the absence of emerin in patients with genetic defects of *EMD* expresses a variant of EDMD with a severe phenotype, and the missense variant expresses a normal or low level of emerin and supports a milder phenotype. The pathogenic variant *LMNA* gene, of the homozygous or compound heterozygous pathogenic form, causes a more severe form of the disease, as in this clinical case [10].

Molecular detection can be determined in *EMD* and *FHL1* pathogenic variants for asymptomatic patients because heterozygous women are at risk of developing heart disorders and muscular dystrophy syndromes [10]. Emery–Dreifuss syndrome can be inherited in either an autosomal dominant (AD-EDMD) or recessive (AR-EDMD) or X-linked (XL-EDMD) manner. Overall, 65% of patients with AD-EDMD have a de novo *LMNA* pathogenic variant. The studies have revealed that each child has a 50% chance of inheriting the pathogenic variant from one of their parents with AD-EDMD [8,9,10].

It cannot be detected by the genetic variants if the disease is AR or AD [9]. In this case, the molecular test performed in the elder sister revealed the same mutation, confirming the autosomal dominant pattern of inheritance.

The clinical symptoms for a diagnosis of EDMD are: limb muscle wasting and/or weakness and elbow or neck/spine joint contractures and a hemizygous pathogenic variant in *EMD* or *FHL1*; a heterozygous pathogenic variant in *LMNA*; or biallelic pathogenic variants in *LMNA* identified by molecular genetic testing [11].

Molecular genetic tests may include a combination of gene-targeted testing (multgene panel or single gene testing) and comprehensive genomic testing (genome sequencing, exome sequencing, or exome arrays), depending on the phenotype.

As the phenotype of EDMD is broad, individuals with the distinctive features described in the Case Presentation above are likely to be diagnosed using gene-targeted testing, whereas those with atypical features in whom the diagnosis of EDMD has not been considered are more likely to be diagnosed using genomic testing [11,12,13].

The evolution of the disease was rapidly progressive, with an unfavorable prognosis: the patient, evaluated cardiologically annually, did not present musculoskeletal or cardiological disorders prior to SARS-CoV2 virus infection. However, a close correlation between COVID-19 infection and the onset or the manifestation of the latent disease cannot be excluded or confirmed.

## 4. Conclusions

In this case, multidisciplinarity was very important, with the patient being diagnosed with Emery–Dreifuss syndrome about 3 months after SARS-CoV2 virus infection, and the severe evolution of the disease required ICD implants at about 4 months after SARS-CoV2 virus infection.

## Figures and Tables

**Figure 1 genes-12-01070-f001:**
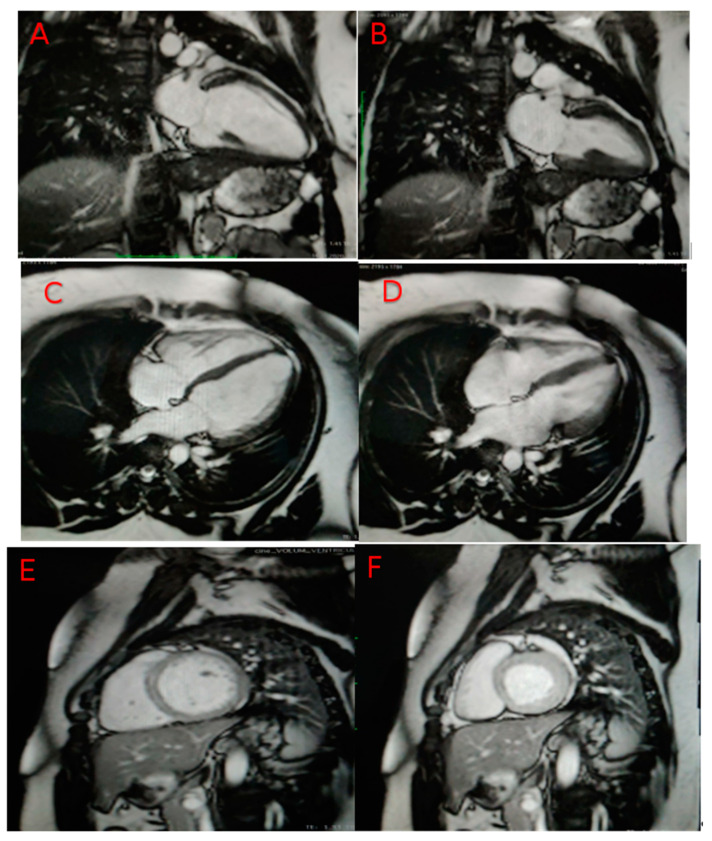
MRI confirms LV dilated cardiomyopathy. In images (**A**) (long axis 2 chambers LV-diastole), (**B**) (long axis 2 chambers LV-systole (**C**) (long axis 4 chambers LV-diastole), (**D**) (long axis 4 chambers LV-systole), (**E**) (short axis LV-diastole), (**F**) (short axis LV-systole), the LV is slightly dilated (end-diastolic volume index-EDVi 98 mL/m^2^—probably underestimated indexed value due to the patient’s weight, with systolic function around the lower limit of normal (LVEF 58%). Ventric-ular volumes: end-diastolic volume (EDV)—220 mL, end-systolic volume (ESV)—92 mL, stroke volume—128 mL, ejec-tion fraction (LVEF: 58%, RVEF: 61%).

**Figure 2 genes-12-01070-f002:**
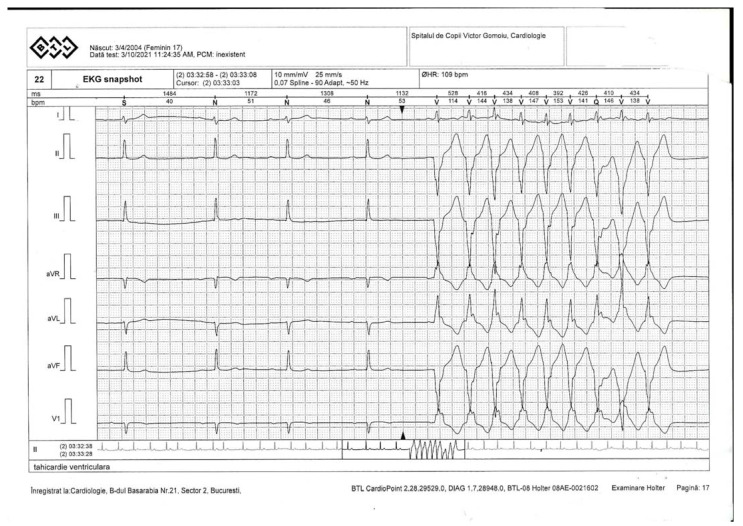
Ventricular tachycardia.

## Data Availability

Restrictions apply to the availability of these data. Data are available from the authors with the permission of the “Alexandru Gafencu” Military Emergency Hospital Constanța.

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
