# Peer review of "SARS-CoV-2 Infection and Emery-Dreifuss Syndrome in a Young Patient with a Family History of Dilated Cardiomyopathy"

_genes, 2021, doi:10.3390/genes12071070_

Round 1

Reviewer 1 Report

The paper needs better organization. Please follow the CARE guidelines for case report.

Author Response

Has been improved English language, results, conclusions, references

I used the CARE guidelines

Reviewer 2 Report

The paper entitled "SARS-CoV-2 infection and Emery-Dreifuss Syndrome in a Young Patient with a Family History of Dilated Cardiomyopathy" by Dumitru et al. is a well-written manuscript and is focused on an actual World Health Problem. SARS-CoV-2 can be acute when other known pathological disorders are presents or even more, came to the light pathological disorders that were unknown before in the patient. It is an easy way to bring up to the reader the clinical effects of SARS-CoV-2.

As minor comments, figures could have a brief explanation of what we are seeing in each panel similar to fig 4, just because for non-clinicians readers could be a bit difficult to understand. 

Author Response

Has been improved English language, Fig.4 deleted

Reviewer 3 Report

Interesting case, but has many issues to solve:

-page 2, line 53: the gene es LAMIN, not LAMININ (that's another gene).

-page 2, line 57: "cardiomyopathy"

-page 2, line 61: ??

-page 2, lines 62-64: update references (it's from 2000), there are several recent articles that must be revised and mentioned: Barriales-Villa R, Rev Esp Cardiol. 2021;74:208-910.1016/j.rec.2020.09.021; Hasselberg NE et al EHJ 2017; Rijsingen et al. J Am Coll Cardiol 2012, Pinto YM et al. Position Statement WG. ESC 2016, Van Berlo et al. J Mol Med (2005) 83:79–83...

-Figure 1: it's not necessary to show. It has no value in this case, mainly because it was normal.

-Figure 2: very bad quality. Low voltages are not mentioned, and that cannot be normal. Give a reason for that.

-Page 4, line 104: what does "lower wall" mean?, use standard nomenclature for regional segments.

-Figure 3: very bad image, must be improved.

-Page 4, line 110: you mention paracetamol as a nonsteroidal anti-inflammatory medication. Check.

-Figure 4: LVEF 58% it's normal. Where is the information about late gadolinium enhancement??, it's mandatory in any cardiac MRI study. How can you diagnose a myocarditis without this information?. This must be addressed to publish the article.

-Page 5, line 130: you must explain the technique used for gene sequencing: NGS?, number of genes studied?.

-Page 6, lines 135-144: summarize all 3 holters in 2-3 lines maximum.

What does "short pauses of NSVT" mean?

-Page 6, lines 147-155: summarize. There is a lot of unnecessary information. Volumes are in mm??, correct.

-Page 6, lines 161-167: Unnecessary information for this case.

-Figure 6: nothing is seen. Null quality.

-Discussion: references 9 and 10 nothing to do with the text.

You don't give any hypothetical theory about the mechanism or relationship between sars-cov2 and myocarditis in lamin mutations.

-Page 7, line 184: "phenotype".

-Page 7, lines 192-193: review literature, lamin mutation are usually inherited from parents, not de novo. Sentence beggining by "Currently...", it has no sense. Please check and correct.

-you must give information about the lamin variant found in the literature. Is it a pathogenic variant?, a probably/possible pathogenic variant?, a VUS?. Add information about how is classified by the ACMG. You also must expand the discussion talking about this variant and other lamin mutations and how do they affect the prognosis of carriers. Also give information about risk factors in lamin carriers.

-Conclusions: it is not appropriate, it does not reflect the whole manuscript. Why you talk about vaccination if you don't mention it in the manuscript?

Author Response

-page 2, line 53: the gene es LAMIN, not LAMININ (that's another gene). lamin A/C

-page 2, line 57: "cardiomyopathy" - cardiomyopathy

-page 2, line 61: ?? - delete ??

-page 2, lines 62-64: update references (it's from 2000), there are several recent articles that must be revised and mentioned: Barriales-Villa R, Rev Esp Cardiol. 2021;74:208-910.1016/j.rec.2020.09.021; Hasselberg NE et al EHJ 2017; Rijsingen et al. J Am Coll Cardiol 2012, Pinto YM et al. Position Statement WG. ESC 2016, Van Berlo et al. J Mol Med (2005) 83:79–83...

  1. – changed references 6 with Barriales-Villa R et al. Risk predictors in a Spanish cohort with cardiac laminopathies. The REDLAMINA registry. Rev Esp Cardiol (Engl Ed). 2021 Mar;74(3):216-224. doi: 10.1016/j.rec.2020.03.026. and Pinto YM, Reckman YJ. Formins Emerge as a Cause of Hypertrophic Cardiomyopathy: New Genes for Thick Hearts. J Am Coll Cardiol. 2018 Nov 13;72(20):2468-2470.

-Figure 1: it's not necessary to show. It has no value in this case, mainly because it was normal. deleted

-Figure 2: very bad quality. Low voltages are not mentioned, and that cannot be normal. Give a reason for that. deleted

-Page 4, line 104: what does "lower wall" mean?, use standard nomenclature for regional segments. I corrected

- Figure 3: very bad image, must be improved. I corrected

-Page 4, line 110: you mention paracetamol as a nonsteroidal anti-inflammatory medication. Check. Deleted nonsteroidal

-Figure 4: LVEF 58% it's normal. Where is the information about late gadolinium enhancement??, it's mandatory in any cardiac MRI study. How can you diagnose a myocarditis without this information?. This must be addressed to publish the article. - I corrected

-Page 5, line 130: you must explain the technique used for gene sequencing: NGS?, number of genes studied?. I corrected

-Page 6, lines 135-144: summarize all 3 holters in 2-3 lines maximum. - I corrected

What does "short pauses of NSVT" mean?

-Page 6, lines 147-155: summarize. There is a lot of unnecessary information. Volumes are in mm??, correct. - I corrected

-Page 6, lines 161-167: Unnecessary information for this case. deleted

-Figure 6: nothing is seen. Null quality. Deleted

-Discussion: references 9 and 10 nothing to do with the text. Deleted,  I put the suggested referencesYou don't give any hypothetical theory about the mechanism or relationship between sars-cov2 and myocarditis in lamin mutations.- deleted

-Page 7, line 184: "phenotype".

-Page 7, lines 192-193: review literature, lamin mutation are usually inherited from parents, not de novo. Sentence beggining by "Currently...", it has no sense. Please check and correct – I corrected

-you must give information about the lamin variant found in the literature. Is it a pathogenic variant?, a probably/possible pathogenic variant?, a VUS?. Add information about how is classified by the ACMG. You also must expand the discussion talking about this variant and other lamin mutations and how do they affect the prognosis of carriers. Also give information about risk factors in lamin carriers. I improved

-Conclusions: it is not appropriate, it does not reflect the whole manuscript. Why you talk about vaccination if you don't mention it in the manuscript? . I improved

Round 2

Reviewer 3 Report

All previous comments has been solved. 

Author Response

Revision 2

- Genes should be indicated in italics; - corrected line 43-44, 53, 54, 56, 68, 73, 144, 182, 184, 191, 198, 199

- The correct abbreviation for the electrocardiogram is ECG, not EKG – corrected, line 39, 84,98, 172

- At line 108, the sentence is incomplete; probably some words are missing - added

- Line 170 "were" instead of "was" - corrected

- The last sentence should be modified as following: "In this case of presentation, it was necessary to carry out a genetic test for EDMD in family members before the ICD implantation. - changed

- Finally a revision of the english language is necessary - receipt MDPI english version 31960

- I added the recommended references

  1. Gujar, H.; Weisenberger, D.J.; Liang, G. The Roles of Human DNA Methyltransferases and Their Isoforms in Shaping the Epigenome. Genes 2019, 10, 172. https://doi.org/10.3390/genes10020172
  2. Motorin, Y.; Helm, M. Methods for RNA Modification Mapping Using Deep Sequencing: Established and New Emerging Technologies. Genes 2019, 10, 35. https://doi.org/10.3390/genes10010035